# Nationally-representative serostudy of dengue in Bangladesh allows generalizable disease burden estimates

Henrik Salje[1,2]*, Kishor Kumar Paul[3], Repon Paul[3], Isabel Rodriguez-Barraquer[4], Ziaur Rahman[3], Mohammad Shafiul Alam[3], Mahmadur Rahman[5], Hasan Mohammad Al-Amin[3], James Heffelfinger[6], Emily Gurley[2]

[1]Mathematical Modelling of Infectious Diseases Unit, Institut Pasteur, UMR2000, CNRS, Paris, France; [2]Department of Epidemiology, Johns Hopkins Bloomberg School of Public Health, Baltimore, United States; [3]International Centre for Diarrhoeal Disease Research, Bangladesh (icddr,b), Dhaka, Bangladesh; [4]University of California, San Francisco, San Francisco, United States; [5]Institute of Epidemiology, Disease Control and Research (IEDCR), Dhaka, Bangladesh; [6]Division of Global Health Protection, Center for Global Health, Centers for Disease Control and Prevention, Atlanta, United States

**Abstract** Serostudies are needed to answer generalizable questions on disease risk. However, recruitment is usually biased by age or location. We present a nationally-representative study for dengue from 70 communities in Bangladesh. We collected data on risk factors, trapped mosquitoes and tested serum for IgG. Out of 5866 individuals, 24% had evidence of historic infection, ranging from 3% in the north to >80% in Dhaka. Being male (aOR:1.8, [95%CI:1.5–2.0]) and recent travel (aOR:1.3, [1.1–1.8]) were linked to seropositivity. We estimate that 40 million [34.3–47.2] people have been infected nationally, with 2.4 million ([1.3–4.5]) annual infections. Had we visited only 20 communities, seropositivity estimates would have ranged from 13% to 37%, highlighting the lack of representativeness generated by small numbers of communities. Our findings have implications for both the design of serosurveys and tackling dengue in Bangladesh.
DOI: https://doi.org/10.7554/eLife.42869.001

*For correspondence:
hsalje@pasteur.fr

**Competing interests:** The authors declare that no competing interests exist.

## Introduction

There has been growing recognition of the utility of nationally representative serum banks to monitor the burden from infectious diseases in a population (*Metcalf et al., 2016*; *Wilson et al., 2012*; *Osborne et al., 1997*; *Jardine et al., 2010*; *De Melker and Conyn-van Spaendonck, 1998*; *van der Klis et al., 2009*). By tracking the levels of pathogen-specific antibodies in populations, these banks are a powerful tool for public health agencies to understand a wide range of factors that can assist in the fight against diseases, including pathogen circulation patterns, vaccination levels, and the existence of spatial pockets of susceptibility (*De Melker and Conyn-van Spaendonck, 1998*; *Gidding et al., 2005*; *Osborne et al., 2000*). To date, the accumulation and use of nationally representative banked sera have focused almost exclusively on vaccine preventable childhood infections. However, serum banks also have the potential to be invaluable in efforts to understand the burden from arboviruses and optimize efforts for control, including the targeted deployment of vaccines (*Imai and Ferguson, 2018*). In these diseases, high levels of subclinical infection and frequent clinical misclassification mean that even in locations with good disease surveillance, the underlying risk of infection is poorly understood (*Halstead, 2007*).

**eLife digest** Dengue is a mosquito-borne virus that infects millions of people each year. Often the countries most affected by the virus, such as Bangladesh, do not have the resources needed to tackle the disease. For resources sent to these countries to have the greatest impact, it is important to know which areas are most affected, and which subsets of the population are most at risk. A way to gather this information is to test for dengue virus antibodies a protein produced by the immune system in response to the infection in the blood of individuals. However, previous efforts to use these tests to understand dengue risk in communities have generally only been done in single locations, typically a major city, and the findings of these tests are unlikely to be applicable to the wider population.

Now, Salje et al. have visited 70 different communities from all around Bangladesh and used these tests on blood samples collected from over 5,000 individuals from a range of age-groups. From these measurements it was estimated that an average 2.4 million people are infected with dengue each year in Bangladesh, with major cities, such as Dhaka, experiencing more concentrated levels. The exposure to dengue outside major cities was much lower, and men, who tend to travel more, were found to be at greater risk of infection.

Salje et al. also showed that using a small number of communities to estimate national levels of infection led to misleading results. This highlights the danger of using information collected from a limited number of places to represent the effects of a disease on the wider population.

Public health agencies in Bangladesh will be able to use this information to tackle dengue more effectively, focusing on the areas and the populations most affected by the disease. In addition, the design and analytical approaches used in this study could be applied to other countries, and to different diseases.

DOI: https://doi.org/10.7554/eLife.42869.002

Studies that collect serum for the detection of pathogen-specific antibodies typically rely on either convenience samples (e.g., blood donations) or focus on single communities, often in major cities or where the perceived risk of the pathogen is believed to be greatest (*Petersen et al., 2012*; *Salje et al., 2016a*; *Rodríguez-Barraquer et al., 2014*). While they are often a good starting place for understanding historic pathogen circulation in the sampled communities and age-groups, the ability to generalize the results to the wider population is rarely known. By contrast, if communities are randomly chosen from across a country, we can estimate national incidence rates, the underlying spatial heterogeneity in burden, the spatial dependence across neighbouring communities and identify risk factors for infection. Such study designs therefore provide mechanistic insights into pathogen spread and facilitate the development of data-informed control policies.

Dengue virus is a flavivirus, transmitted by *Aedes* mosquitoes, that is found across tropical and subtropical regions and causes a range of disease manifestations, ranging from asymptomatic infection to death (*Petersen et al., 2012*). Transmission of arboviruses, such as dengue, appears to be driven by the interplay of individual- (e.g., sex, age, travel), household- (e.g., water supply, use of mosquito control) and community-level (e.g., urban/rural, mosquito abundance) factors (*Salje et al., 2016b*; *Rodríguez-Barraquer et al., 2015*). In order to make data-informed decisions on how best to control spread, we need to understand the relative importance of these different factors by collecting detailed data across these scales. A recent literature search found only one nationally representative dengue seroprevalence study, from Singapore, but there was only a subset of age-groups considered (*Imai et al., 2015*).

Outside of city states such as Singapore, Bangladesh is the most densely populated country in the world with 146 million people living in an area under 150,000 km$^2$. The dengue burden in Bangladesh is unclear. Sporadic cases were reported in the 1960 s and a major outbreak occurred in 2000 (*Rahman et al., 2002*; *Sharmin et al., 2015*; *Yunus et al., 2001*), with clinical cases reported annually since then (*Government of the People's Republic of Bangladesh, Ministry of health and family Welfare, 2017*). However, our knowledge of dengue epidemiology in the country is largely restricted to Dhaka, where a seroprevalence of 80% has been observed (*Dhar-Chowdhury et al., 2017*), with the burden elsewhere unknown (*Government of the People's Republic of Bangladesh,*

*Ministry of health and family Welfare, 2017*). Here, we present the results of a study where we use sequential annual visits in randomly selected communities across Bangladesh to determine the burden of dengue and identify key risk factors for infection.

## Materials and methods

### Community and household selection

We randomly selected 70 communities from the 97,162 communities in the national census, where the probability of selection was proportional to the size of the community population. In rural locations (around three-quarters of the country), these census-communities consist of villages, whereas in urban places, these communities are city wards. Study teams visited each of the selected communities at least twice, once during the period 08/2014-12/2014 (Y1) and once during the period 10/2015-01/2016 (Y2) to conduct interviews, collect serum and trap mosquitoes. A further visit was conducted in 06/2015-07/2015 in a subset of communities for additional mosquito collection only. For each visit, the study team spent at least 5 days in the community. In an attempt to select households randomly, the study staff identified the house where the most recent wedding had taken place and identified the closest neighbour. They then counted six households in a random direction to identify the first household for the study. To select each additional household for the study, they used the previous household as a starting point and counted six households in a random direction. Different households were selected in each visit. For selected households, the household head was informed of the study and invited to participate. If the household head was away during the first visit, the study team returned at a later time. If the household head agreed to participate, all household residents over the age of 6 months were also invited to participate. Residents were offered a test to determine their blood group as a benefit of participating. If some members of the household agreed and some refused, all consenting members were included in the study. Where some household members were not present at the time of the visit, study staff organised a time to come back. Data collection for a community was considered to be complete when at least 40 serum samples from at least 10 households had been collected. There were three elements to data collection: (A) questionnaires (B) serum collection and (C) mosquito collection.

### Questionnaires

Each participant was led through a questionnaire. Where individuals were too young to answer, older individuals from the household answered for them. We asked a series of questions on demographics (age, sex), whether they had ever been diagnosed with dengue and whether they had travelled outside of their community in the prior 7 days, 30 days or 6 months. In addition, the head of the household was asked to complete a separate questionnaire, which included questions about their education level, total household income, household utilities (e.g., access to electricity and clean water), whether they had used any form of mosquito control in the last week and whether they owned land away from the household.

### Serum sample collection and testing

A phlebotomist collected 5 ml of venous blood from all individuals who gave consent. Individuals who were sick at the time were ineligible. These samples were centrifuged in the field and the serum extracted into separate vials before being shipped to icddr,b (previously known as the international centre for diarrhoeal disease research, Bangladesh) laboratories in Dhaka in nitrogen dry shippers. The samples were tested for antibodies against IgG dengue virus, which indicates historic infection, using PanBio indirect IgG ELISAs (Alere Inc, Massachusetts, USA).

### Mosquito collection

During the first visit in 2014, BG Sentinel traps (Biogents AG, Germany) were placed in eight randomly sampled households in each of the of the 70 participating communities. The traps were placed in the main living area of the households and after 24 hr, they were collected and all mosquitoes sent to icddr,b laboratories in Dhaka where an entomologist identified the species of each captured mosquito. To help ensure that communities where no *Aedes* mosquitoes were found during the initial mosquito trapping truly had no *Aedes*, mosquito trapping was repeated in these

communities from June 2015 to July 2015 during which eight households were randomly selected to have sentinel traps placed in their homes for 24 hr and the traps and mosquitos were processed in the same way as they had been initially.

## Understanding risk factors for infection using regression analysis

We divided the covariates into individual-level (age, sex, travel patterns), household-level (household income, electricity in household, access to water in household, mosquito control) and community-level (*Ae. albopictus/Ae. aegypti* in community, log population size) categories. For each covariate, we initially performed simple logistic regression to explore associations with dengue serostatus using a hierarchical model with random intercepts for household and community. We accounted for spatial correlation structure using a Matern covariance function using a stochastic partial differential equation and fitted the models using integrated nested Laplace approximations (INLA) in a Bayesian framework (*Lindgren et al., 2011*). All covariates were then included in a multivariable analysis. As the probability of being seropositive is strongly linked to the past circulation of dengue, we also performed a sensitivity model in which we recalculated the regressions using only individuals > 20 years as seropositivity was not found to differ by age among older adults. Finally, we assessed the importance of using spatial correlation structure by calculating the coefficients in a separate regression that did not include a spatial covariance matrix.

## Mapping the risk of dengue across Bangladesh

To explore the variability in dengue risk across Bangladesh, we initially placed a 5 km x 5 km grid over the country and estimated the population size in each of those grid cells using data available from worldpop.org (*Tatem, 2017*). We then fit a multivariable model using the data from our sampled locations using log(population size), age category (<10 y, 11-20y, 21-30y, 31-40y, 41y-50, 51-60y,>60 y) and sex as covariates, which represent variables that are either available for all the grid cells (population size) or where we can use the overall proportion of the population that is within each category (age and sex) from the national census. As above, we fit the model in a Bayesian framework with a Matern spatial correlation structure using integrated nested Laplace approximations. We used the fitted model to predict in the unsampled grid cells by drawing 1000 samples from the posterior for each grid cell and calculated the mean as well as 2.5% and 97.5% quantiles to quantify uncertainty. The estimated number of seropositive individuals in a cell was calculated by multiplying the estimated proportion seropositive in a cell by the population within that cell. The total number of seropositive individuals in the country was calculated as the sum of seropositive individuals across all the grid cells. As a sensitivity analysis, we also predicted the spatial distribution of dengue seropositivity in the country using a model with the Matern spatial covariance matrix only (i. e., without any covariates).

We assessed the ability of different model formulations to accurately predict the level of seropositivity in unsampled locations. We considered four different models: (i) crude proportion seropositive, (ii) multivariable logistic model with sex, age-group and population size as covariates and Matern spatial covariance (the baseline model), (iii) multivariable logistic model with sex, age-group and population size as covariates but with no spatial dependence, and (iv) spatial dependence model using Matern spatial covariance with no covariates. For 100 iterations and for each model in turn, we repeatedly randomly selected a subset of communities to train the model (varied between 2, 20, 40, 60 and 69 communities) and predicted the seroprevalence in the remaining communities not used to fit the model. Separately, we considered the impact of having sampled fewer people per community. We reran each of the four models over repeated iterations using 50 randomly selected communities and with between 2 and 80 individuals sampled per community to train the models. We then estimated the seroprevalence in the 20 remaining communities.

## Estimation of the force of infection using catalytic models

We used the probability of being seropositive as a function of age to estimate the proportion of the susceptible population that get infected each year using catalytic models, an approach which has been used frequently to reconstruct the past circulation of pathogens (*Salje et al., 2016a*; *Rodríguez-Barraquer et al., 2014*; *Imai et al., 2015*; *Ferguson et al., 1999*). We assumed a constant force of infection due to all four serotypes, λ and that there were no differences in risk by age.

The proportion seropositive of age $a$, is given by $z(a)=1-exp(-\lambda \times (min(a,NYears))$, where *NYears* is the number of years prior to 2014 that dengue has circulated in Bangladesh. We fixed *NYears* at 20 to reflect the approximate period when dengue first appeared in the country. We conducted a sensitivity analysis where this was varied between 15 and 25 years. We estimated $\lambda$ using maximum likelihood where the contribution to the likelihood from seronegative individuals coming from community $i$ is $exp(-\lambda \times (min(a,NYears)) \times 1/wt(comm_i)$, where the weights, $wt(comm_i)$, represent the proportion of that community that was sampled (number of people in community i/population in community i). This approach was used to ensure that all individuals contributed equally to the likelihood. The contribution to the likelihood from seropositive individuals is $(1-exp(-\lambda \times (min(a,NYears))) \times 1/wt(comm_i)$.

We calculated the force of infection for the entire sampled population as well as separate estimates by sex and for the locations from the three largest cities (Dhaka, Chittagong and Khulna) only versus the rest of the country.

## Estimation of the number of infected individuals per year

To estimate the number of people infected each year we used the estimated population by age for each year for the period 1995–2014 (*Kinsella and He, 2009*). We assumed that in 1995, the entire population was susceptible. The proportion of the population that have monotypic immunity is calculated as $w(a,y)=4 \times exp(-3 \times \lambda_s \times a^*) \times (1-exp(-\lambda_s \times a^*))$ where $\lambda_s$ is the serotype specific force of infection and is calculated as $\lambda/4$ and $a^*$ is the number of years an individual has been alive since the introduction and is calculated as $min(a,y-1995)$. Similarly, the proportion of the population that has previously been infected with two serotypes ($w2(a,y)$) is $6 \times exp(-2 \times \lambda_s \times a^*) \times (1-exp(-\lambda_s \times a^*))^2$ and the number previously infected with three serotypes $w3(a,y)=4 \times exp(-\lambda_s \times a^*) \times (1-exp(-\lambda_s \times a^*))^3$. Using these proportions we can calculate the number of primary, secondary, tertiary and quaternary dengue infections. Where $N(a,y) \times 4 \times \lambda_s * exp(-4 \times \lambda_s \times a^*)$ is the number of primary infections, $N(a,y) \times 3 \times \lambda_s \times w(a,y)$, the number of secondary infections, $N(a,y) \times 2 \times \lambda_s \times w2(a,y)$ the number of tertiary infections and $N(a,y) \times \lambda_s \times w3(a,y)$ the number of quaternary infections. $N(a,y)$ is the size of the population of age group $a$ in year $y$. We present the estimated total number of infections across primary, secondary, tertiary and quaternary infections.

## Ethical review

This study was approved by the icddr,b ethical review board (protocol number PR-14058). The U.S. Centers for Disease Control and Prevention relied on icddr,b's ethical review board approval. All adult participants provided written, informed consent after receiving detailed explanation of the study and procedures. Parents/guardians of all child participants were asked to provide written, informed consent on their behalf.

## Results

In total, 5866 individuals fully participated (completed questionnaire and had blood taken) in our study across 70 communities, 2911 during August–December 2014 and 2955 during October 2015 – January 2016 (*Table 1*). We obtained serum from 76% of household members of participating households (*Figure 1—figure supplement 1*). Per community there were an average of 95 participants (range 81–116) from 20 households (range 20–23). The age and sex distributions in the study largely matched those obtained by the 2011 census although we had some under-representation in those <10 years (*Figure 1—figure supplement 2*). The PanBio assay appeared to discriminate well between those with and without past dengue infection (*Figure 1—figure supplement 3*). We found that overall, 24% of individuals had evidence of a past infection. We observed substantial heterogeneity by age and sex (*Figure 1B*), with 27% of males seropositive compared to 21% of females (p-value<0.001). Individuals > 20 y had 30% seropositivity compared to 14% in those under 20y. There was close correlation between the proportion seropositive in a community between 2014 and 2015 (Pearson correlation of 0.92) (*Figure 1C*). Overall, there was no observed difference in seropositivity across the two years of the study (p=0.66). While most of the study population (91%) aged >10 y had heard of dengue, only 38 individuals (0.6%) reported having had dengue, of whom only 16 had evidence of past infection.

While all communities had at least one seropositive individual, there was substantial spatial heterogeneity across the country with the proportion seropositive ranging from 3% in rural Maulvibazar

**Table 1.** Individual-, household- and community-level characteristics of participants, stratified by serostatus to dengue.

| | Serum obtained (N = 5,866) | Seropositive (N = 1,403) | Seronegative (N = 4,463) |
|---|---|---|---|
| Individual level | n | n (%) | n (%) |
| Year of study | | | |
| 2014 | 2911 | 704 (24) | 2207 (76) |
| 2015 | 2955 | 699 (24) | 2256 (76) |
| Age group (years) in 2014: | | | |
| <10 | 832 | 88 (11) | 744 (89) |
| 11–20 | 1062 | 314 (30) | 748 (70) |
| 21–30 | 1402 | 228 (16) | 1174 (84) |
| 31–40 | 818 | 261 (32) | 557 (68) |
| 41–50 | 679 | 197 (29) | 482 (71) |
| 51–60 | 541 | 144 (27) | 397 (73) |
| >60 | 525 | 171 (33) | 354 (67) |
| Sex: | | | |
| Male | 2821 | 761 (27) | 2060 (73) |
| Female | 3044 | 642 (21) | 2402 (79) |
| Heard of dengue: | | | |
| No | 772 | 115 (15) | 657 (85) |
| Yes | 2093 | 1288 (25) | 3805 (75) |
| Reported having had dengue: | | | |
| No | 5827 | 1387 (24) | 4440 (76) |
| Yes | 38 | 16 (42) | 22 (58) |
| Last time left community: | | | |
| <7 days | 773 | 269 (35) | 504 (65) |
| 7d-1 month | 1198 | 311 (26) | 887 (74) |
| 1–6 months | 1142 | 285 (25) | 857 (75) |
| >6 months | 2753 | 538 (20) | 2215 (80) |
| Household level | | | |
| Electricity in home | | | |
| No | 780 | 149 (19) | 631 (81) |
| Yes | 5075 | 1253 (25) | 3822 (75) |
| Access to water in home* | | | |
| No | 301 | 125 (42) | 176 (58) |
| Yes | 2599 | 578 (22) | 2021 (78) |
| Own home | | | |
| No | 444 | 239 (54) | 205 (46) |
| Yes | 5408 | 1161 (21) | 4247 (79) |
| Own land away from home | | | |
| No | 1303 | 373 (29) | 930(71) |
| Yes | 4552 | 1029 (23) | 3523 (77) |
| Mosquito control used | | | |
| No | 2185 | 512 (23) | 1673 (77) |
| Yes | 3670 | 890 (24) | 2780 (76) |
| Household head education: | | | |

*Table 1 continued on next page*

*Table 1 continued*

| | Serum obtained (N = 5,866) | Seropositive (N = 1,403) | Seronegative (N = 4,463) |
|---|---|---|---|
| No education | 1034 | 306 (30) | 728 (70) |
| Primary school | 1574 | 362 (23) | 1212 (77) |
| High school | 1407 | 320 (23) | 1087 (77) |
| Higher | 1840 | 414 (22) | 1426 (78) |
| Household income (Taka, 100 Taka = 1.2 USD): | | | |
| <7000 | 921 | 212 (23) | 709 (77) |
| 7,000–9999 | 1176 | 229 (19) | 947 (81) |
| 10000–20,000 | 1980 | 509 (26) | 1471 (74) |
| >20,000 | 1766 | 452 (26) | 1314 (74) |
| Community level | | | |
| *Aedes aegypti mosquitos captured* | | | |
| No | 3931 | 668 (17) | 3263 (83) |
| Yes | 1935 | 735 (38) | 1200 (62) |
| *Aedes albopictus mosquitos captured* | | | |
| No | 3416 | 941 (28) | 2475 (72) |
| Yes | 2450 | 462 (19) | 1988 (81) |
| Type of community | | | |
| Urban | 1505 | 557 (37) | 948 (63) |
| Rural | 4361 | 846 (19) | 3515 (81) |
| Division: | | | |
| Dhaka | 1484 | 407 (27) | 1077 (73) |
| Chittagong | 1533 | 382 (25) | 1151 (75) |
| Barisal | 329 | 68 (21) | 261 (79) |
| Khulna | 672 | 302 (45) | 370 (55) |
| Rajshahi | 668 | 152 (23) | 516 (77) |
| Rangpur | 920 | 80 (9) | 840 (91) |
| Sylhet | 260 | 12 (5) | 248 (95) |

DOI: https://doi.org/10.7554/eLife.42869.007

in Sylhet Division to 88% in urban Chittagong. Communities in the north of the country appeared largely unaffected. Communities in the northern division of Rangpur had a mean seropositivity of 9% compared to 45% for communities in Khulna division in the southeast. Even within Dhaka district (which includes the capital and has the highest population density), where we visited three urban ('Thana') communities, there was substantial heterogeneity, with seropositivity ranging from 36 to 85%. The two urban communities we visited in the city of Chittagong had seropositivities of 84 and 88%.

We found that several individual-level variables were associated with seropositivity (*Table 2*). In particular, males were much more likely to be seropositive (odds ratio [OR]: 1.6 [95%CI: 1.4–1.9]; adjusted odds ratio [aOR]: 1.7 [1.5–2.0]), although this difference was concentrated in communities where overall seropositivity was <20% (*Figure 2—figure supplement 1*). Travel also appeared important, with those who had travelled in the prior 7 days having twice the odds of being seropositive compared to those that had not travelled in the prior 6 months (OR: 1.9 [1.5–2.4]; aOR: 1.4 [1.1–1.8]). Household-level covariates did not appear to be important in determining risk of seropositivity, including having household electricity, household access to clean water, land ownership or household income. The use of mosquito control in the household was also not associated with

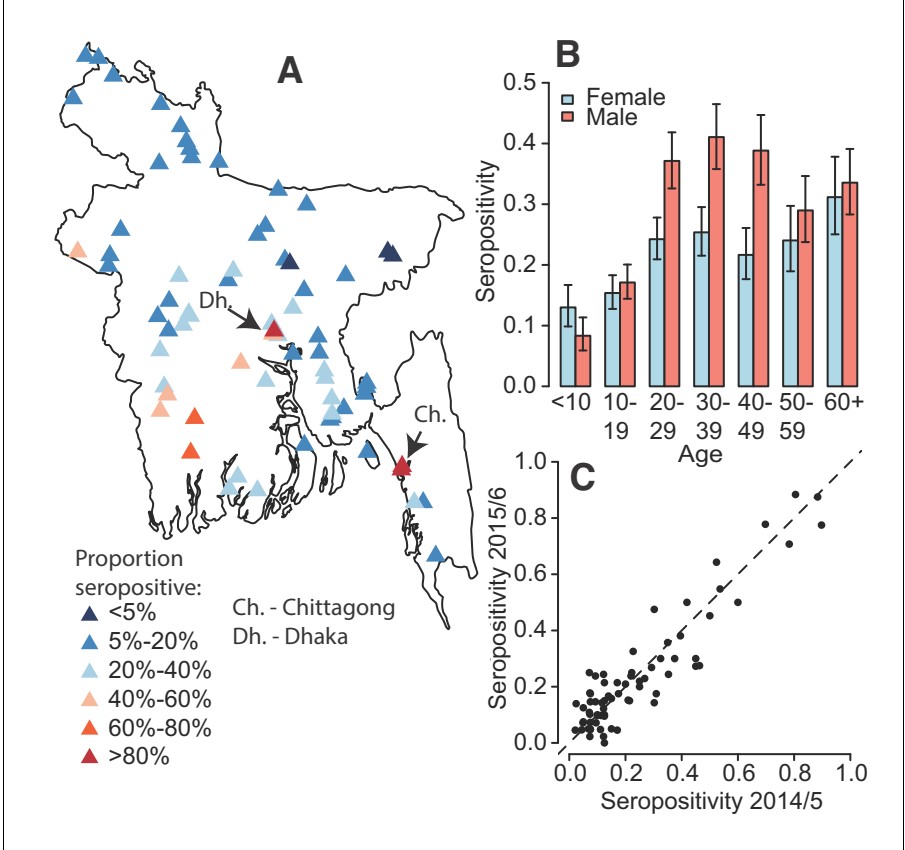

**Figure 1.** Dengue seropositivity in the sampled communities. (**A**) Locations of sampled communities and the estimated seroprevalence by community. (**B**) Proportion seropositive by age and sex with 95% confidence intervals. (**C**) Seropositivity in Y1 compared to seropositivity in Y2 for each community.

DOI: https://doi.org/10.7554/eLife.42869.003

The following figure supplements are available for figure 1:

**Figure supplement 1.** Participants included in study.

DOI: https://doi.org/10.7554/eLife.42869.004

**Figure supplement 2.** Distribution of age groups in years and sex (solid bars) in the study population compared to the 2011 census (lines).

DOI: https://doi.org/10.7554/eLife.42869.005

**Figure supplement 3.** Histogram of PanBio Units derived from the optical densities of the PanBio assay with the manufacturer recommended cutpoint (PanBio Unit of 11) in dashed red.

DOI: https://doi.org/10.7554/eLife.42869.006

seropositivity (OR: 0.9 [0.8–1.1]; aOR: 0.9 [0.7–1.1]). At the community-level, we found some evidence that individuals living in locations where we had found *Ae. aegypti* were more likely to be seropositive, although this effect was less in the multivariable model (OR 1.8 [1.2–2.8], aOR 1.4 [0.9–2.2]). Having *Ae. albopictus* in the community was not linked to individual serostatus (OR 1.1 [0.7–1.6], aOR 1.0 [0.7–1.6]). Overall *Ae. aegypti* was found in 23 (33%) and *Ae. albopictus* in 29 (41%) of communities with a slight negative correlation between the two (Pearson correlation of −0.2, p-value 0.07). The median seropositivity in communities with *Ae. aegypti* was 33% compared to 13% in the other communities. The median seropositivity in communities with *Ae. albopictus* was 15% compared to 18% in communities where it was not found. Individuals living in urban communities were more likely to be seropositive than those living in rural communities with each unit increase in log population size associated with a 1.3 times increased probability of being seropositive (95% CI: 1.2–1.5). The intraclass correlation coefficients showed that the Matern spatial covariance matrix explained 15% of the variance, the community-level random effects explained 6% and the household random intercept explained 12% of the variance in individual level responses. In a model without the

**Table 2.** Regression results.

| | Unadjusted | Multivariable |
|---|---|---|
| **Individual level** | Odds ratio (95% confidence interval) | Adjusted odds ratio (95% confidence interval) |
| Year of study (vs Y1) | 1.0 (0.8–1.1) | 1.0 (0.9–1.2) |
| Age group (years) in 2014: | | |
| <10 | Ref | Ref |
| 11–20 | 1.9 (1.4–2.6) | 2.0 (1.4–2.7) |
| 21–30 | 5.2 (3.8–7.1) | 5.5 (4.1–7.6) |
| 31–40 | 5.9 (4.3–8.1) | 6.2 (4.5–8.6) |
| 41–50 | 5.5 (4.0–7.7) | 5.8 (4.2–8.2) |
| 51–60 | 5.1 (3.6–7.1) | 5.1 (3.6–7.2) |
| >60 | 7.5 (5.4–10.6) | 7.7 (5.4–10.8) |
| Male | 1.6 (1.4–1.9) | 1.7 (1.5–2.0) |
| Last time left community: | | |
| <7 days | 1.9 (1.5–2.4) | 1.4 (1.1–1.8) |
| 7d-1 month | 1.4 (1.2–1.7) | 1.2 (0.9–1.4) |
| 1–6 months | 1.1 (0.9–1.3) | 1.0 (0.8–1.2) |
| >6 months | Ref | Ref |
| **Household level** | | |
| Electricity in home | 1.0 (0.8–1.2) | 0.9 (0.7–1.2) |
| Water in home | 1.0 (0.7–1.4) | - (1) |
| Own home | 0.9 (0.7–1.3) | 0.9 (0.7–1.4) |
| Own land | 0.9 (0.8–1.1) | 0.9 (0.7–1.1) |
| Mosquito control used | 0.9 (0.8–1.1) | 0.9 (0.7–1.1) |
| Household head education: | | |
| No education | Ref | Ref |
| Primary school | 0.9 (0.7–1.1) | 0.9 (0.7–1.1) |
| High school | 0.9 (0.7–1.1) | 1.0 (0.7–1.2) |
| Higher | 0.8 (0.7–1.0) | 0.8 (0.6–1.0) |
| Household income (Taka): | | |
| <7000 | Ref | Ref |
| 7,000–9999 | 0.9 (0.7–1.2) | 0.9 (0.7–1.1) |
| 10000–20,000 | 1.0 (0.8–1.2) | 1.0 (0.7–1.2) |
| >20,000 | 0.8 (0.6–1.0) | 0.8 (0.6–1.0) |
| **Community level** | | |
| Population density (log scale) | 1.3 (1.2–1.5) | 1.3 (1.1–1.8) |
| *Aedes aegypti* mosquitos captured | 1.8 (1.2–2.8) | 1.4 (0.9–2.2) |
| Aedes albopictus mosquitos captured | 1.1 (0.7–1.6) | 1.0 (0.7–1.6) |

DOI: https://doi.org/10.7554/eLife.42869.008

spatial covariance matrix, the community-level random intercept explained 23% of the variance with the household-level effect unchanged. Including spatial covariance was associated with a small improvement in model fit, justifying its inclusion (Deviance Information Criterion [DIC] difference of 4). While most of the coefficient estimates were largely consistent in models that did and did not include the spatial covariance structure, the impact of Ae. *aegypti* changed significantly, increasing to aOR 2.4 (95% CI: 1.3–4.5) when spatial correlation was not incorporated (*Figure 2—figure supplement 2*). Not including random intercepts by household and community resulted in falsely narrow confidence intervals and some changes in coefficient estimates and a substantial drop in model fit

(DIC difference of 801). Coefficients of models where the data was restricted to adults only were largely unchanged.

We used a spatial prediction model that incorporates the population size and sex distribution and spatial correlation structure to estimate the level of seropositivity throughout the country (*Figure 2A*). We found that the proportion of people seropositive in communities was spatially correlated up to 108 km (as measured from the Matern covariance function), consistent with that observed in a variogram of the seropositivity between communities (*Figure 2B*). Our model performed well at estimating the observed levels of seropositivity in participating communities in leave-one-out cross-validation with a Pearson correlation of 0.8 between the observed and fitted values and a mean absolute error of 8% (*Figure 2C*). These maps further suggest dengue is currently concentrated in the three largest cities of Dhaka, Chittagong and Khulna. This estimated distribution of dengue risk in the country was very similar if we used the spatial dependence information only, without age and sex covariates (*Figure 2—figure supplement 3*). Overall, we estimate that approximately 25% (95% CI: 21–29%) of the population had been infected with dengue at some point during their lives, equivalent to 40.3 million individuals (95% CI: 34.3–47.2). This estimate is consistent with that obtained using the crude proportion seropositive among our samples (24% or 39.0 million individuals).

Using a catalytic model to estimate the proportion seropositive by age, we estimated that 1.6% (95% CI: 1.5–1.7%) of the susceptible population gets infected each year across the four serotypes, equivalent to an average of 2.4 million annual infections (95% CI: 2.2–2.5 million) (*Figure 2—figure supplement 4*). However, estimates were much higher for the three major urban hubs of Dhaka, Chittagong and Khulna compared to the rest of the country. Within these hubs, 6.4% (95% CI: 5.4–7.6%) of the population gets infected annually with no differences by sex, whereas this drops to 1.0% (95% CI: 0.9–1.2%) for females outside these areas and 1.6% (95% 1.4%–1.8%) for males (*Figure 2D–E*).

We assessed the sensitivity of our results to the number of participating communities and the model framework used. Over repeated iterations, we used a subset of our communities to estimate the overall proportion seropositive and to train a suite of models that were then used to estimate seropositivity in the remaining communities. We found that if we had only visited twenty communities, the seropositivity among the samples would have ranged from 13% to 37%, depending on the communities visited. Spatially explicit models which incorporated data on sex, age and population size, did not result in substantial improvements with the range of seropositivity estimates similarly wide (*Figure 3A*). By contrast, had we visited 60 communities, the range would have been much smaller (22–28%), with similar results in the spatial models. The accuracy of our predictions in unsampled locations improved substantially with increasing numbers of communities visited (*Figure 3B*). In spatial models with no covariates, the mean absolute error in the predictions per community fell from 13.6% when 20 communities were sampled to 10.5% when 69 communities were sampled, with a corresponding rise in the correlation between the observed and predicted seroprevalence from 0.39 to 0.81 (*Figure 3C*). Incorporating information on age, sex and population size resulted in a small improvement in performance when between 20 and 60 communities were sampled (e.g., when 20 communities were sampled, the mean correlation was 0.39 when no covariates were used and 0.49 when covariates were incorporated). Multivariable models with age, sex and population size as covariates but with no spatial correlation performed poorly, with no improvements with increasing numbers of communities visited. We found that sampling fewer people per community had little effect on our estimates, with the performance of nationwide and community-level seropositivity similar if 20 people were sampled per location compared to 80 (*Figure 3D–F*).

## Discussion

We have presented the results of a large, nationally-representative, serostudy that provides a comprehensive description of dengue infection in Bangladesh. Our results demonstrate that, to date, dengue risk is very heterogeneous across the country. It also shows that the vast majority of the country has never been infected. The framework presented here can act as a strategy for future efforts to estimate nationally-representative infection risks in a population.

Our results suggest that since dengue re-emerged in the late 1990 s, the virus has only established a pattern of sustained endemic transmission in a few urban settings, and not throughout the

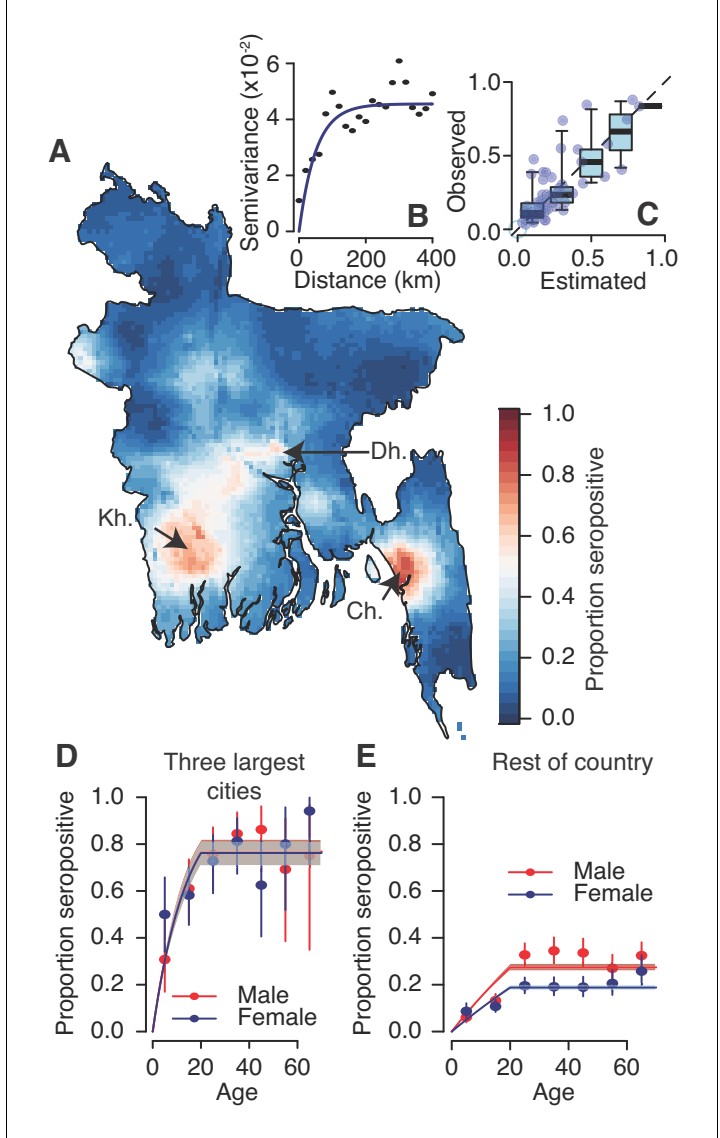

**Figure 2.** Modelled dengue seropositivity across Bangladesh and across age groups. (A) Spatial predictions of seropositivity for the whole country. Kh. = Khulna, Dh. = Dhaka, Ch. = Chittagong (B) Semivariogram showing spatial dependence between the proportion seropositive between communities as a function of distance between them. (C) Observed versus predicted levels of seropositivity by community from leave one out cross validation. (D) Observed (points) and fitted seropositivity by age for the sampled communities within the three largest cities (Khulna, Chittagong and Dhaka) for both males and females. (E) Observed and fitted seropositivity for the remaining communities by sex.

DOI: https://doi.org/10.7554/eLife.42869.009

The following figure supplements are available for figure 2:

**Figure supplement 1.** Relative risk of being seropositive for males versus females as a function of the overall proportion seropositive in the community.

DOI: https://doi.org/10.7554/eLife.42869.010

**Figure supplement 2.** Differences in coefficient estimates in multivariable models run using logistic regression with no random intercept or spatial covariance (blue), with random intercepts at the household and community level (red) and with random intercepts at the household and community level and a Matern spatial covariance matrix (black, base model).

DOI: https://doi.org/10.7554/eLife.42869.011

**Figure supplement 3.** Comparison of risk maps using different prediction methods.

DOI: https://doi.org/10.7554/eLife.42869.012

*Figure 2 continued on next page*

*Figure 2 continued*

**Figure supplement 4.** Observed proportion seropositive by age group with 95% confidence intervals (black) and the fit using the force of infection estimated by the catalytic model (red).

DOI: https://doi.org/10.7554/eLife.42869.013

country, as is the case in nearby Myanmar, Thailand and Cambodia (*Rahman et al., 2002*; *Government of the People's Republic of Bangladesh, Ministry of health and family Welfare, 2017*; *van Panhuis et al., 2015*). Part of the reason for this may be the limited presence of the principal vector, *Ae. aegypti*, which was found in only one third of the communities that participated in this study. By contrast, more (especially rural) communities had the secondary vector, *Ae. albopictus*, but its presence was not associated with infection. Our finding of a negative correlation between *Ae. aegypti* and *Ae. albopictus* presence is consistent with the species occupying different environmental niches or competition between the two species, as has previously been suggested (*Braks et al., 2004*). All communities had at least one seropositive individual, suggesting that external viral introductions may be common and that there may be factors preventing large-scale

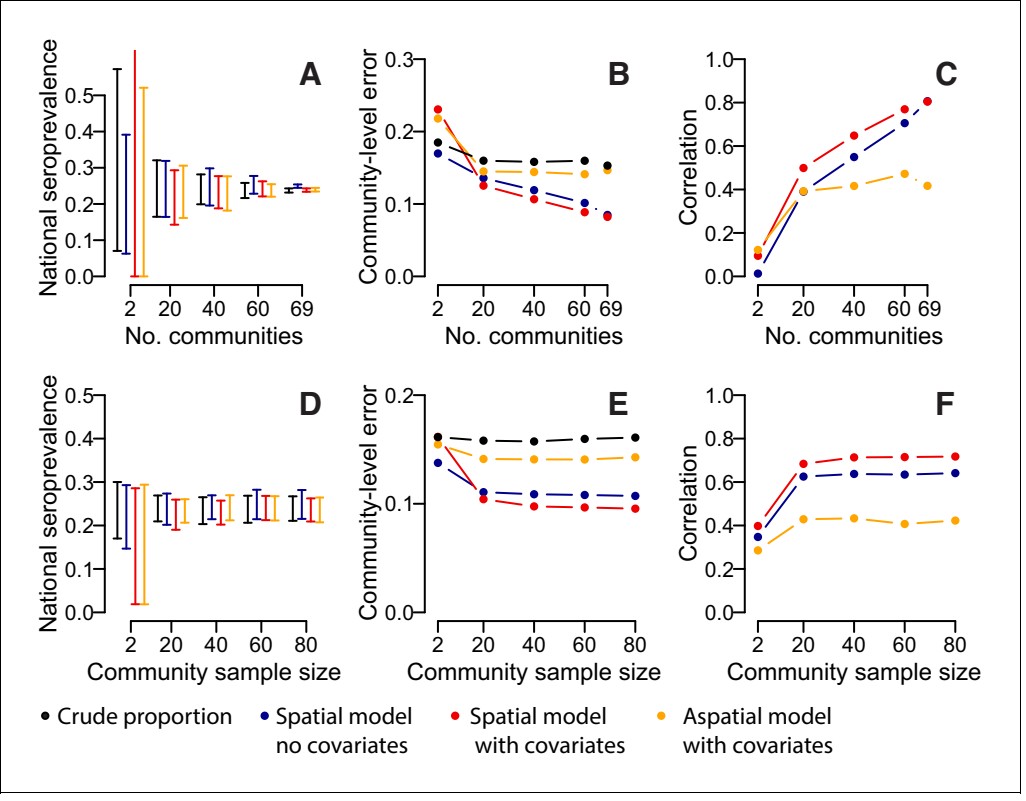

**Figure 3.** Accuracy of estimates for different number of sampled communities (top row) and different numbers of sampled individuals per community (bottom row) using different estimation methods. (**A**) 95% range of estimates of overall seroprevalence from 100 repeated iterations when data from a random subset of communities is used. (**B**) Mean absolute error among heldout communities over repeated iterations. (**C**) Mean correlation between predicted and observed seroprevalence among heldout communities. (**D**) 95% range of estimates of overall seroprevalence from 100 repeated iterations when data from a random number of individuals from 50 communities is used. (**E**) Mean absolute error among 20 randomly selected heldout communities over repeated iterations. (**F**) Mean correlation between predicted and observed seroprevalence among 20 randomly selected heldout communities. The different estimation methods are overall proportion seropositive (black), spatial correlation model using Matern covariance structure and no covariates (blue), spatial correlation model using Matern covariance structure and age, sex and population size as covariates, logistic regression using age, sex and population size as covariates with no spatial component (orange).

DOI: https://doi.org/10.7554/eLife.42869.014

outbreaks, including from the limited abundance of the *Ae. aegypti* vector. It is unclear how stable these vector populations are. Characterizing the drivers of *Ae. aegypti* spread and maintenance, especially in the context of changing land use and climate, appears key to understanding future risk of spread.

We were able to use our study to identify risk factors for infection. At the individual level, we found that males had 1.6 times the odds of having been infected as females, although this difference was concentrated in communities where overall seropositivity was low, suggesting that comparing infection proportions between males and females can be a good marker of local dengue endemicity. In addition, those that travelled more (as indicated by having left the community recently) were more likely to have been infected. These findings suggest that in the current scenario in which infection risk is heterogeneously distributed around the country, individuals from communities with low or no transmission are likely to get infected when they travel to higher transmission areas. If the vector presence expands throughout the country, these individuals could act as sources of outbreaks within these communities. Our findings are in marked contrast to what has been observed with chikungunya in Bangladesh, where women in a community that had a widespread chikungunya epidemic were found to be at significantly increased risk of being infected compared to males, with the increased risk of infection linked to greater time spent in and around the home (*Salje et al., 2016b*). These findings suggest that it may be difficult to generalize inferences across arboviruses due to differences in vector species and the frequency of introductions and risk of onwards spread. While we incorporated spatial correlation into our risk factor regression analyses, in practice this only resulted in a relatively small improvement in model fit compared to hierarchical models with random intercepts at the community and household levels. The biggest impact of incorporating the spatial correlation was to move the coefficient estimate for Ae. *aegypti* presence towards the null. This suggests that the covariance structure is a better predictor than the basic mosquito absence/presence data as the covariance is driven by, and absorbs, the true underlying environmental drivers including mosquito distributions.

Overall, we estimated that around a quarter of the population, around 40 million individuals, have been infected by dengue with an average of 2.4 million annual infections. This figure is much less than the 16.7 million previously estimated through a modelling exercise (*Bhatt et al., 2013*), highlighting the need for representative data to help support these models and the importance of considering immunity in the estimation of annual case numbers. We did not detect a significant difference in seropositivity between the two study years, though we were not sufficiently powered to detect small changes in seropositivity across the population. This points to a clear trade-off between resampling the same individuals across the two study years, which would have facilitated quantification of the incidence between the two years and our approach, which allowed us to maximize our sample size. While there was substantial spatial heterogeneity in the risk of being seropositive across the country, ultimately our crude estimates of the proportion seropositive in the country (24%) was very close to modelled estimates that incorporated the age, sex and population distribution in the country (25%). This provides strong support for the sampling frame we used to capture population-level estimates of population exposure.

While spatial prediction models did not help improve overall estimates of national burden, they did allow us to build maps of how infection risk is distributed throughout the country. Incorporating covariates (e.g., sex, age, population distribution) did not result in substantive improvements in predictive accuracy compared to models that included a spatial covariance term only. This highlights the importance of spatial dependence in obtaining accurate estimates in unsampled locations. The use of environmental and climate covariates were not considered here and may improve estimates, especially where insufficient (or no) sampled locations exist to make use of spatial correlation structure. Our approach is particularly relevant to settings like Bangladesh where there was little prior understanding of the distribution in disease burden in the country. Alternative strategies may exist in other settings where there is already some existing knowledge of where risk is concentrated. Although, even in these settings, unmeasured spatial differences in healthcare seeking or in surveillance system infrastructure may mean that it is preferable to randomly sample communities without specifically focusing on specific areas of populations in the country.

Our study provides key insight for the national vaccine policy for dengue. For the only currently licensed vaccine, Dengvaxia, the WHO has recommended that countries perform nationally representative serosurveys to inform vaccine rollout as the vaccine only provides protection in people

with existing antibodies (*World Health Organization, 2016*). The vaccination of seronegative individuals has been linked to increased risk of subsequent severe disease (*Salje et al., 2018*; *Hadinegoro et al., 2015*). For Bangladesh, our findings suggest that any vaccine rollout should be concentrated to the urban areas of Dhaka, Chittagong and Khulna. However, even in these communities, the proportion seropositive at age 9 years is far below the threshold of 80% where vaccine rollout is potentially feasible without pre-vaccination screening (*World Health Organization, 2018*). Therefore, any rollout will require the screening of individuals for presence of antibodies before vaccination to avoid placing large numbers of individuals at risk for more severe disease manifestations.

The approach presented here could be used as a strategy for other countries interested in obtaining national estimates of disease risk. The optimal number of communities to visit will depend on the size and distribution of the population, the underlying level and heterogeneity of infection in the population, the required level of precision and the available budget. Therefore, it is difficult to make general recommendations about the number of communities which should be sampled in other settings. However, our finding that spatial correlation exists within 100 km suggests that communities should be sampled at a density to ensure that there is at least one sampled community within 100 km of all residents and preferably more. Sampling as few as 20 individuals per community still provides robust nationwide estimates, however, in practice, there are fewer budget and time constraints to sampling additional individuals within a community than visiting additional communities.

Cross-reactivity of antibodies is a problem for all seroprevalence studies, especially with flaviviruses. This prevents us from quantifying the relative importance of the different dengue serotypes. In addition, some seropositive individuals may have been infected with Japanese encephalitis rather than dengue. However, Japanese encephalitis is typically only found in rural communities where it circulates at low levels (estimated at 2.7 cases/100,000) (*Paul et al., 2011*). As we estimated only low levels of dengue seropositivity in rural communities, the number of false positives from Japanese encephalitis cross-reactivity is likely to be small. Individuals who participated in the study may not be representative of all members in the community. In particular, individuals who travel frequently may have been away. We attempted to minimize this risk by organizing times to meet with household members who were not present in the initial visit. Around 90% of households that we approached agreed to take part in the study. We used BG sentinel traps that have been shown to be well suited to trapping *Aedes* mosquitoes (*Maciel-de-Freitas et al., 2006*; *Obenauer et al., 2010*). In addition, we revisited all communities where we did not find *Aedes* mosquitoes in the initial visit for additional mosquito trapping. However, given the heterogeneous nature of mosquito distributions within communities, we may have nevertheless failed to find *Aedes* mosquitoes in communities where they breed. This would mean that *Aedes* may be more widespread than we found. We used the time period since individuals last left the community as a marker of travel. While this is likely to broadly capture trends in mobility, it remains a crude marker and more detailed measures of movement (from e.g., movement diaries, global positioning system monitors) would help provide a more detailed understanding of how people move. To randomly select households in a community, we would ideally have used a sampling frame of all households in the community. However, in this setting there were no detailed community maps and enumerating all households in the communities would have added an additional day in the field per community. Therefore, we used a quasi-random approach that identified a starting point for household sampling based on the area of the community where the most recent wedding took place; given that >95% of adults marry in Bangladesh, it is unlikely that this approach could bias the sample we obtained. However, in cultural contexts where marriage rates may vary by community or location within a community, this method of choosing a random starting point could produce a biased sample.

We found that simply asking people about whether or not they had been infected with dengue was not informative of past infection. This highlights that studies that only use questionnaires can only provide limited burden information for pathogens such as dengue. For example, Demographic Health Surveys, which collect detailed health questionnaires from randomly selected individuals across many countries could benefit significantly from adding a serological component to their studies. In Bangladesh, where dengue is still emerging, surveillance for vectors could be a way to monitor risk of future outbreaks and continued efforts to understand drivers of transmission could point to interventions to reduce its geographic spread.

## Additional information

### Funding

| Funder | Author |
| --- | --- |
| Centers for Disease Control and Prevention | Henrik Salje Emily Gurley |

The funders had no role in study design, data collection and interpretation, or the decision to submit the work for publication.

### Author contributions

Henrik Salje, Conceptualization, Formal analysis, Supervision, Funding acquisition, Validation, Investigation, Visualization, Methodology, Writing—original draft, Project administration; Kishor Kumar Paul, Data curation, Supervision, Investigation, Project administration, Writing—review and editing; Repon Paul, James Heffelfinger, Project administration, Writing—review and editing; Isabel Rodriguez-Barraquer, Formal analysis, Methodology, Writing—review and editing; Ziaur Rahman, Hasan Mohammad Al-Amin, Supervision, Investigation, Methodology, Writing—review and editing; Mohammad Shafiul Alam, Investigation, Methodology, Writing—review and editing; Mahmadur Rahman, Supervision, Investigation, Project administration, Writing—review and editing; Emily Gurley, Conceptualization, Formal analysis, Supervision, Funding acquisition, Investigation, Methodology, Project administration, Writing—review and editing

### Author ORCIDs

Henrik Salje http://orcid.org/0000-0003-3626-4254
Kishor Kumar Paul http://orcid.org/0000-0002-6054-3571
Isabel Rodriguez-Barraquer http://orcid.org/0000-0001-6784-1021
Emily Gurley http://orcid.org/0000-0002-8648-9403

### Ethics

Human subjects: This study was approved by the icddr,b ethical review board. (protocol number PR-14058). The U.S. Centers for Disease Control and Prevention relied on icddr,b's ethical review board approval. All adult participants provided written, informed consent after receiving detailed explanation of the study and procedures. Parents/guardians of all child participants were asked to provide written, informed consent on their behalf.

### Decision letter and Author response

Decision letter https://doi.org/10.7554/eLife.42869.019
Author response https://doi.org/10.7554/eLife.42869.020

## Additional files

### Supplementary files

• Source data 1. Dengue data.
DOI: https://doi.org/10.7554/eLife.42869.015
• Transparent reporting form
DOI: https://doi.org/10.7554/eLife.42869.016

### Data availability

All data generated or analysed during this study are included in the manuscript and supporting files with the exception of precise coordinate and age information.

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
