## [Decision Letter]

Thank you for submitting your article "Nationally-representative serostudies are needed for generalizable burden estimates: dengue in Bangladesh case-study" for consideration by *eLife*. Your article has been reviewed by three peer reviewers, including Ben Cooper as the Reviewing Editor and Reviewer #1, and the evaluation has been overseen by Neil Ferguson as the Senior Editor. The following individual involved in review of your submission has agreed to reveal their identity: Oliver Brady (Reviewer #3).

The reviewers have discussed the reviews with one another and the Reviewing Editor has drafted this decision to help you prepare a revised submission.

Summary:

Nationally representative serum samples are a valuable tool for understanding disease dynamics and planning vaccination interventions, but outside high income countries such studies are lacking (what seroprevalence studies there are are typically based on convenience samples). The current study, which reports the results of a nationally representative serological study for of Dengue in Bangladesh provides an important insight into the burden of disease and informs public health and future intervention strategies. It also represents an important demonstration of how such a national serological study can be conducted and analysed in a LMIC setting and how it can inform policy.

Essential revisions:

There was a consensus amongst reviewers that the analysis does need a bit more work – certainly in description and possibly also in rigour. At the moment there do seem to be missing some detail as outlined below and it is unclear how the risk factor analysis links to the burden predictions.

1) The title and Abstract of the manuscript are very heavily focused on the need to conduct a large number (> 10) seroprevalence surveys to generate accurate estimates of burden, but this is not tested in detail in the paper. Aside from the numerous other methods of estimating dengue burden which is probably outside of the scope of this paper, the analyses presented here only focus on randomised sampling schemes. WHO guidance on conducting seroprevalence surveys for dengue published only last year (Informing vaccination programs: a guide to the design and conduct of dengue serosurveys, WHO, Geneva, 2017) recommends stratified sampling based on historic dengue incidence as documented by passive surveillance. Passive surveillance data may not be available in Bangladesh, but it is in the vast majority of dengue endemic countries, thus limiting the generalizability of the findings presented here. Using the spatial correlation or risk factor analysis might also give unbiased estimates from < 10 well placed surveys (not tested here). It was also difficult to reconcile these interpretations with statements like: "ultimately our crude estimates of the proportion seropositive in the country (24%) was very close to an estimate adjusted for the age and sex distribution in the country". There is real value in the work that has been done here in its own right, so it is a little confusing why the focus of the paper is so heavily skewed towards a statement that was not so thoroughly tested.

2) Using leave one out cross validation in a dataset of size 70 communities is probably not a particularly stringent test. At a minimum, a CV split of 80% training 20% testing would be more standard. Looking at the model fit (2A-B) even with this small hold-out set there seems to be a fair amount of unexplained variance. Could this be better explained by including other (non-spatial) covariates – ideally those related to the risk factors identified in Table 2? Would this change the conclusions about minimum site sample size for national representativeness?

3) How confident are the authors of the claim made in the second paragraph of the Introduction that nationally representative serum samples are lacking outside high income countries? Can this be backed up by a systematic search of the literature, or is there robust evidence from one or more of the cited papers (Metcalf et al., 2016; Wilson et al., 2012; Osborne, Weinberg and Miller, 1997; Jardine et al., 2010; De Melker and Conyn-van Spaendonck 1998; Ang et al., 2015)?

4) The analysis accounts for clustering at the village level (through a hierarchical model) but not spatial correlation, which in general would be expected to reduce the amount of information in the data. It would be useful if this could be discussed, the decision to ignore such correlation justified, and the situations where it is important to account for spatial correlation in the regression modelling discussed.

5) Results section: "no household level covariates were significantly linked to seropositivity". Following the ASA's report on p-values (and many recommendations before that), it is widely considered unwise to report results according to "bright lines" for p-values (in this case 0.05 presumably). By all means report the p-values, and certainly report the confidence interval, but hopefully we are moving away from the era where we use such arbitrary thresholds to decide whether to report results, while ignoring the magnitude of the effect.

6) It is slightly difficult to read this paper as the Materials and methods are at the end and not enough detail is given in the Results for it be clear what was actually done without referring to the Materials and methods. Note that *eLife* guidelines say that "A Methods or Model section can appear after the Introduction where it makes sense to do so". I think the authors should either consider moving the Materials and methods to before the Results or at least briefly saying what was done in the Results with a more detailed explanation in the Materials and methods.

7) It's quite unclear how the results were used to derive nationally representatives estimates. The Materials and methods simply say "we also calculated a census-adjusted proportion seropositive by community that adjusted for any sampling bias" but details of how this was done are lacking. Such adjustment also seems to have been using only age and sex distribution from the 2011 census. Why not also use Urban vs. rural (an important predictor according to Table 2 and which should be easily obtained). Other community-level covariates could also be used, potentially. Note there is a literature on multilevel regression and post-stratification to address this type of problem that might be relevant here (see, for example, Zhang et al. Am J Epidemiol 2014, https://academic.oup.com/aje/article/179/8/1025/109078).

8) References to Figure 2B and Figure 2C in the text need to be swapped.

9) "All covariates that were statistically significant at a p-value of <0.1 in the unadjusted analysis were included in a multivariable analysis." Not clear why this was done (except that many other papers do this). There seem to be enough data to include all covariates, and covariates with p-values >0.1 may still have important affects either alone or in combinations. Sometimes covariates have to be selected using some approach as there aren't enough data to use them all, but this doesn't seem to be the case here.

10) "using catalytic models". Given there is no space limit it would be helpful to define the technical details here. Also, a constant force of infection was assumed for all serotypes. Was this assumption of a constant f.o.i. tested (e.g. by comparison with other models)? How consistent are the data with this assumption? The Materials and methods suggest that f.o.i. was allowed to vary by age (is this correct)? Why not also sex given the reported sex differences?

11) "we estimate that 40 million people have been infected with 2.3 million annual infections." (Abstract and Results). 95% CIs are needed for these estimates?

12) Sample size calculations will be useful to others. However, those currently given in the supplementary text use a formula without further justification (or reference to any justification) and condition on the fact that 70 communities are being sampled. To be more useful, I think it would be help if the authors could discuss the sample size implications of sampling more (or fewer) communities for given correlations between village. For example, for the observed within-village correlation how would the required sample size change as a function of the number of communities sampled. What would be the impact of spatial correlation be on these numbers? What about different levels of within village correlation? This would be useful for others planning such studies and think it would also be appropriate to discuss this issue in the Discussion section i.e. what are the resources required for a given precision/resolution and how are the resources required likely affected by study design choices. It would also be useful if the authors could at least discuss the issue of spatial correlation and discuss the merits/demerits of accounting for it in the modelling in general (as well as in this specific example).

13) "Reported having dengue" and "heard of dengue" seem peculiar things to include in the regression, as these might be extended to be a consequence of having dengue rather than a potential risk factor. Perhaps this reflects the fact that the purpose of the regression analysis is not clearly stated. It would be good to have a clear statement about the purpose of the regression and a justification for the choice of variables to include.

14) The supplementary material describes the recruitment strategy: "the study staff identified the house where the most recent wedding had taken place and identified the closest neighbour. They then counted six households in a random direction to identify the first household for the study. To select each additional household for the study, they used the previous household as a starting point and counted six households in a random direction." This seems a little eccentric and not obviously guaranteed to give a random sample. Is there any justification for this choice? Wouldn't numbering households and selecting at random or throwing darts at maps be better? I think this point at least needs to be discussed (with recommendations for future studies) and this aspect of the methods should be move to the Materials and methods section in the main text.

15) "may provide some guidance". Not sure what the intended meaning of this is. What kind of guidance?

16) Was any attempt made at assessing the accuracy of the recorded household data?

17) The entomological approach to determining the presence or absence of *Ae. aegypti* is quite superficial. The intensity of surveillance (number of BG traps per community) and duration (time in the field) is too short to arrive at a conclusion of presence/absence. In dengue endemic cities, where *Ae. aegypti* has a well-documented presence, there can be quite marked spatial heterogeneity in the distribution of *Ae. aegypti* when measured by BG traps, e.g. some houses can be free of this species for consecutive weeks but in a house 50 metres away they can be caught regularly. In regards to *Ae. albopictus* prevalence, BG traps set indoors are not the optimal method of determining presence/absence- it would have better to set them outdoors or to use outdoor ovitraps. The authors should qualify their conclusions by recognising that trapping method, intensity, duration and seasonality can all influence the likelihood of Aedes detection and this could change the conclusions of the manuscript.

18) Is there any reason uncertainty (in either or all of the data, kriging model and the force of infection model) can't be propagated through to the final burden estimates? Comparing mean estimates with Bhatt et al. to prove that nationally representative surveys are needed probably also needs to consider uncertainty. I think they might have also included tertiary and quaternary infections as well if you want to be comparable.

19) Can code for statistical analysis be made available?

20) "Our findings are in marked contrast to what has been observed with chikungunya in Bangladesh". Can the authors offer any hypothesis as to why this might be the case?

21). Can the authors provide at least one concrete example of such a survey could lead to better decision-making about vaccination?

22) It was felt that some of the statements about findings showing that lack of spread of *aegypti* is the reason behind heterogeneities in dengue transmission in Bangladesh were not fully justified in light of the known limitations of entomological surveying.

23) The authors correctly point to the possibility that the Panbio-based seroprevalence survey might have reflected past JEV exposure and cite the low case JE incidence to suggest dengue is the primary culprit for the seroloprevalence.. This might be true, but JEV is notoriously difficult to diagnose in the absence of laboratory testing and there would almost certainly be under-reporting of cases in Bangladesh. Having a random subset of samples tested by DENV/JEV PRNT50 assay, regarded as the most specific assay of DENV serostatus, could have helped clarify this point. Can the authors do this readily perhaps on a sample of 100 or so (or use Luminex as a (less preferred) option? Though not essential, it was felt this would improve the quality of the manuscript if it could be done within 2 months. If it can't be done the Discussion needs to be qualified accordingly.

24) Materials and methods: when were interviews conducted in relation to the dengue season? Could this have introduced recall bias of "whether diagnosed with dengue"?

25) Was there any evidence that travel in the last 7 days was a reasonable proxy for long term travel history?

26) "Our finding of a negative correlation between *Ae. aegypti* and *Ae. albopictus* presence is consistent with competition between the two species" – or is just evidence that they have different environmental niches?

---

## [Author Response]

Essential revisions:There was a consensus amongst reviewers that the analysis does need a bit more work – certainly in description and possibly also in rigour. At the moment there do seem to be missing some detail as outlined below and it is unclear how the risk factor analysis links to the burden predictions.1) The title and Abstract of the manuscript are very heavily focused on the need to conduct a large number (> 10) seroprevalence surveys to generate accurate estimates of burden, but this is not tested in detail in the paper. Aside from the numerous other methods of estimating dengue burden which is probably outside of the scope of this paper, the analyses presented here only focus on randomised sampling schemes. WHO guidance on conducting seroprevalence surveys for dengue published only last year (Informing vaccination programs: a guide to the design and conduct of dengue serosurveys, WHO, Geneva, 2017) recommends stratified sampling based on historic dengue incidence as documented by passive surveillance. Passive surveillance data may not be available in Bangladesh, but it is in the vast majority of dengue endemic countries, thus limiting the generalizability of the findings presented here. Using the spatial correlation or risk factor analysis might also give unbiased estimates from < 10 well placed surveys (not tested here). It was also difficult to reconcile these interpretations with statements like: "ultimately our crude estimates of the proportion seropositive in the country (24%) was very close to an estimate adjusted for the age and sex distribution in the country". There is real value in the work that has been done here in its own right, so it is a little confusing why the focus of the paper is so heavily skewed towards a statement that was not so thoroughly tested.

We thank the reviewer and editor on their thoughts and agree that in some settings there exist sufficient data that could guide alternative sampling. strategies. The optimal sampling strategy is these other settings is certainly an interesting question but we believe falls outside the scope of this paper. We now discuss that alternative strategies may exist where some knowledge already exists. We, however, disagree with the statement that “Passive surveillance data may not be available in Bangladesh, but it is in the vast majority of dengue endemic countries”. For example, India, the country which (probably) suffers from the greatest disease burden from dengue, the situation is very poorly understood with little/no epidemiological data from large (especially rural) areas. Similarly, there is little dengue epi data from African countries – a setting where dengue case reports are frequent but there is rarely systematic surveillance. Further, even in countries with developed surveillance systems like Thailand, only a minority of cases are laboratory confirmed (in Thailand this has been estimated at 10%) and are therefore based on clinical presentation only, which results in frequent misdiagnosis. There are also important differences in healthcare seeking behaviours across the country. Therefore there may be an important discrepancy between what is recorded in surveillance systems and true underlying burden. With regards to the WHO guidance on conducting seroprevalence studies – some of the coauthors (along with the reviewer) were involved in drafting the guidance document. It is very specifically focused on dengue vaccine guidance rather than understanding the level of seropositivity. As such, it e.g., specifically does not consider areas where few cases are recorded (which would be the majority of Bangladesh) and only focuses on children. Therefore it is only of limited relevance to this study.

We agree with the reviewers that in our first draft we did not robustly compare the possible inferences from different analytical approaches and the number of communities/people sampled as indicated by the title/Abstract. Therefore in the revised manuscript we have now systematically compared the nationwide and community level inferences using different approaches and assessed the accuracy of estimates with different numbers of communities and participants per community. The different analytical approaches are:

1) Crude proportion seropositive

2) Spatial covariance with no covariates

3) Spatial covariance with age, sex and population size covariates (this is our baseline model, following suggestions by the reviewer)

4) Age, sex, population size covariates and no spatial covariance

As set out in a new Figure (Figure 3), we find that for the overall nationwide estimate, there is little difference in the uncertainty of estimates across the methods – i.e., incorporating a spatial model and/or covariates does not improve estimates when only a small number of places are sampled. So for example, that if 20 communities had been chosen out of our 70 – the overall crude estimate would have ranged between 12% and 37% depending on which communities were chosen with the range of values using spatial models being very similar. For community-level estimates in held out locations (i.e., the goal of spatial prediction exercises), we find that interestingly, adding covariates provides only minor improvement in predictive accuracy compared to a basic spatial correlation model – this appears to be because the covariates themselves (population size) are themselves spatially correlated and therefore, their effect can be absorbed in the covariance matrix.

In the revised document, we describe the different model formulations in the Materials and methods. We include a new figure (Figure 3). In the Materials and methods we include: “We assessed the ability of different model formulations to accurately predict the level of seropositivity in unsampled locations. […] We then estimated the seroprevalence in the 20 remaining communities.”

In the Results we include:

“We assessed the sensitivity of our results to the number of participating communities and the model framework used. […] We found that sampling fewer people per community would have had little effect on our estimates, with the performance of nationwide and community-level seropositivity similar if 20 people were sampled per location compared to 80 (Figures 3D-F).”

In the Discussion, we now include:

“While spatial prediction models did not help improve overall estimates of national burden, they did allow us to build maps of how infection risk is distributed throughout the country. […] Although, even in these settings, unmeasured spatial differences in healthcare seeking or in surveillance system infrastructure may mean that it is preferable to randomly sample communities without specifically focusing on specific areas of populations in the country.”

2) Using leave one out cross validation in a dataset of size 70 communities is probably not a particularly stringent test. At a minimum, a CV split of 80% training 20% testing would be more standard. Looking at the model fit (2A-B) even with this small hold-out set there seems to be a fair amount of unexplained variance. Could this be better explained by including other (non-spatial) covariates – ideally those related to the risk factors identified in Table 2? Would this change the conclusions about minimum site sample size for national representativeness?

While we agree that in many analyses, 20% held out testing makes more sense – here, the main information being used is the spatial location of the sampled units and their associated community seropositivity. We are therefore asking the question – can we estimate the seroprevalence at an unsampled location using the spatial covariance from sampled locations (i.e., the information is obtained in the spatial covariance matrix – which goes away if there are no sampled communities nearby). Leaving out 20% changes the question to “if we had sampled 56 locations, what are the estimates obtained”. Therefore we prefer to keep the leave one out analysis. However, we thank the reviewer for this comment as it motivated us to explicitly explore the impact of only having observed a subset of communities (which is equivalent to varying the proportion in the training set and the proportion in the validation set as suggested by the reviewer) and then applying different spatial prediction models. In the revised manuscript, we assess the different models to accurately estimate seroprevalence when between 2 and 69 communities are used to train the model and the rest used to validate the model.

Separately, we also agree that risk factors could potentially improve predictions. Following the reviewer’s suggestion, we now include population size and age/sex in the spatial prediction maps. Note that in Bangladesh, there do not exist maps that can link individual communities (and associated census data) to lat/lon points. We believe that this complication is not unique to Bangladesh with communities in e.g., India and African countries also having similar problems.

We now estimate the predictive accuracy of four different approaches under different numbers of communities and participants per community (Figure 3).

We also include predictive maps with population size, age and sex as spatial predictors as the main analysis (Figure 2—figure supplement 3A) but also compare to a map using spatial covariance only (Figure 2—figure supplement 3B).

3) How confident are the authors of the claim made in the second paragraph of the Introduction that nationally representative serum samples are lacking outside high income countries? Can this be backed up by a systematic search of the literature, or is there robust evidence from one or more of the cited papers (Metcalf et al., 2016; Wilson et al., 2012; Osborne, Weinberg and Miller, 1997; Jardine et al., 2010; De Melker and Conyn-van Spaendonck 1998; Ang et al., 2015)?

We agree that this statement was overly broad and difficult to back up. We have therefore removed this sentence.

4) The analysis accounts for clustering at the village level (through a hierarchical model) but not spatial correlation, which in general would be expected to reduce the amount of information in the data. It would be useful if this could be discussed, the decision to ignore such correlation justified, and the situations where it is important to account for spatial correlation in the regression modelling discussed.

This is an interesting point that warranted further investigation. In order to explore the impact of the estimation method, we have moved to a Bayesian framework. In the revised manuscript we compare the inferences made when using (1) logistic regression with no random intercepts (2) random intercepts by household/community but without spatial correlation (i.e., what was in the original submission) and (3) where we also include spatial dependence through a spatial fields approach as implemented in INLA (which has now become our baseline model). We find that the latter model is slightly better supported (DIC difference of 4). Interestingly, the point estimations for the individual and household covariates are very similar for the models 2 and 3. However, the impact of the mosquito populations do go towards the null. This is likely as a result in the strong spatial correlation in the mosquito populations themselves that gets absorbed in the spatial covariance matrix. We include a discussion of these analyses in the revised document and a new figure showing the different in coefficients (Figure 2—figure supplement 2).

We now include a comparison of coefficient estimates under different model formulations (Figure 2—figure supplement 2). We include in the Results:

“The intraclass correlation coefficients showed that the Matern spatial covariance matrix explained 15% of the variance, the community-level random effects explained 6% and the household random intercept explained 12% of the variance in individual level responses. [...] Coefficients of models were the data was restricted to adults only were largely unchanged.”

In the Discussion we include: “While we incorporated spatial correlation into our risk factor regression analyses, in practice this only resulted in a relatively small improvement in model fit compared to hierarchical models with random intercepts at the community and household levels. […] This suggests that the covariance structure is a better predictor than the basic mosquito absence/presence data as the covariance is driven by, and absorbs, the true underlying environmental drivers including mosquito distributions.”

5) Results section "no household level covariates were significantly linked to seropositivity". Following the ASA's report on p-values (and many recommendations before that), it is widely considered unwise to report results according to "bright lines" for p-values (in this case 0.05 presumably). By all means report the p-values, and certainly report the confidence interval, but hopefully we are moving away from the era where we use such arbitrary thresholds to decide whether to report results, while ignoring the magnitude of the effect.

We included this language as it has become standard in the field, however, we agree completely that this is a shame and we should avoid it where possible. We have reworded this phrase in the revised document to “Household-level covariates did not appear to be important in determining risk of seropositivity”.

6) It is slightly difficult to read this paper as the Materials and methods are at the end and not enough detail is given in the Results for it be clear what was actually done without referring to the Materials and methods. Note that eLife guidelines say that "A Methods or Model section can appear after the Introduction where it makes sense to do so". I think the authors should either consider moving the Materials and methods to before the Results or at least briefly saying what was done in the Results with a more detailed explanation in the Materials and methods.

As suggested, we have moved the Materials and methods to after the Introduction.

7) It's quite unclear how the results were used to derive nationally representatives estimates. The Materials and methods simply say "we also calculated a census-adjusted proportion seropositive by community that adjusted for any sampling bias" but details of how this was done are lacking. Such adjustment also seems to have been using only age and sex distribution from the 2011 census. Why not also use Urban vs. rural (an important predictor according to Table 2 and which should be easily obtained). Other community-level covariates could also be used, potentially. Note there is a literature on multilevel regression and post-stratification to address this type of problem that might be relevant here (see, for example, Zhang et al. Am J Epidemiol 2014, https://academic.oup.com/aje/article/179/8/1025/109078).

Following the reviewer’s suggestions (here and elsewhere), we have moved the spatial predictions to a Bayesian framework and incorporate sex, age, population size (values that are easily available for the rest of the country). We also compared the performance of different model formulations. We also have added detail into the Materials and methods as to how we did the spatial prediction.

We have changed the main analysis to include population size, sex and age as suggested. We include comparisons of maps that do and do not include these covariates (Figure 2—figure supplement 4) and also the predictive accuracy of these different formulations (Figure 3).

In the Materials and methods we now include “To explore the variability in dengue risk across Bangladesh, we initially placed a 5km x 5km grid over the country and estimated the population size in each of those grid cells using data available from worldpop.org 23. […] As a sensitivity analysis, we also predicted the spatial distribution of dengue seropositivity in the country using a model with the Matern spatial covariance matrix only (i.e., without any covariates).”

8) References to Figure 2B and Figure 2C in the text need to be swapped.

Figure 2 has now changed and the references have been checked.

9) "All covariates that were statistically significant at a p-value of <0.1 in the unadjusted analysis were included in a multivariable analysis." Not clear why this was done (except that many other papers do this). There seem to be enough data to include all covariates, and covariates with p-values >0.1 may still have important affects either alone or in combinations. Sometimes covariates have to be selected using some approach as there aren't enough data to use them all, but this doesn't seem to be the case here.

As suggested, we now include all covariates in the multivariable model (the results are essentially unchanged).

10) "using catalytic models". Given there is no space limit it would be helpful to define the technical details here. Also, a constant force of infection was assumed for all serotypes. Was this assumption of a constant f.o.i. tested (e.g. by comparison with other models)? How consistent are the data with this assumption? The Materials and methods suggest that f.o.i. was allowed to vary by age (is this correct)? Why not also sex given the reported sex differences?

We have now included the technical details of the model. We thank the reviewer for the suggestions and we also run models for the main urban sites and by sex. We find that they highlight the key differences in the force of infection in these different populations. We include a new figure highlighting this (Figure 2D-E). We assume no differences in risk by age – this has been clarified.

We include a new section in the Materials and methods “Estimation of the force of infection using catalytic models” which sets out the likelihood based approach for calculating the force of infection:

“Estimation of the force of infection using catalytic models

We used the probability of being seropositive as a function of age to estimate the proportion of the susceptible population that get infected each year using catalytic models, an approach which has been used frequently to reconstruct the past circulation of pathogens 12, 13, 16, 24. […] We calculated the force of infection for the entire sampled population as well as separate estimates by sex and for the locations from the three largest cities (Dhaka, Chittagong and Khulna) only versus the rest of the country.”

In the Results we include a new figure (Figure 2D-E) with the force of infection estimate by the different populations.

We also include the text: “Using a catalytic model to estimate the proportion seropositive by age, we estimated that 1.6% (95% CI: 1.5%-1.7%) of the susceptible population gets infected each year across the four serotypes, equivalent to an average of 2.4 million annual infections (95% CI: 2.2-2.5 million) (Figure 2—figure supplement 4). However, estimates were much higher for the three major urban hubs of Dhaka, Chittagong and Khulna compared to the rest of the country. Within these hubs, 6.4% (95% CI: 5.4%7.6%) of the population gets infected annually with no differences by sex, whereas this drops to 1.0% (95% CI: 0.9% – 1.2%) for females outside these areas and 1.6% (95% 1.4% -1.8%) for males (Figure 2D, E).”

11) "we estimate that 40 million people have been infected with 2.3 million annual infections." (Abstract and Results). 95% CIs are needed for these estimates?

We now include confidence intervals for these estimates.

12) Sample size calculations will be useful to others. However, those currently given in the supplementary text use a formula without further justification (or reference to any justification) and condition on the fact that 70 communities are being sampled. To be more useful, I think it would be help if the authors could discuss the sample size implications of sampling more (or fewer) communities for given correlations between village. For example, for the observed within-village correlation how would the required sample size change as a function of the number of communities sampled. What would be the impact of spatial correlation be on these numbers? What about different levels of within village correlation? This would be useful for others planning such studies and think it would also be appropriate to discuss this issue in the Discussion section i.e. what are the resources required for a given precision/resolution and how are the resources required likely affected by study design choices. It would also be useful if the authors could at least discuss the issue of spatial correlation and discuss the merits/demerits of accounting for it in the modelling in general (as well as in this specific example).

We thank the reviewer for the comment – we agree that sample size considerations are important. The optimal number of communities to visit will depend on the size and distribution of the population, the underlying level and heterogeneity of infection in the population, the required level of precision and the available budget and whether the goal is to measure overall exposure or have community-specific estimates. We did not feel we could do justice to all these considerations in a simple sample-size guide. Instead, we feel the overall findings of the paper will help future studies. We also now provide the impact of within household and within community correlation and a provide rough example guidance in the Discussion.

With regards to the issue of accounting for spatial correlation, we now include analyses where we explicitly consider the impact of not including it – both on coefficient estimates (Figure 2—figure supplement 2) and how it helps with spatial prediction (Figure 3).

In the Results we include the estimates of the intraclass correlation coefficients. In the Discussion we include:

“The approach presented here could be used as a strategy for other countries interested in obtaining national estimates of disease risk. […] Sampling as few as 20 individuals per community still provides robust nationwide estimates, however, in practice, there are fewer budget and time constraints to sampling additional individuals within a community than visiting additional communities.”

We also include “While we incorporated spatial correlation into our risk factor regression analyses, in practice this only resulted in a relatively small improvement in model fit compared to hierarchical models with random intercepts at the community and household levels. […] This suggests that the covariance structure is a better predictor than the basic mosquito absence/presence data as the covariance is driven by, and absorbs, the true underlying environmental drivers including mosquito distributions.”

13) "Reported having dengue" and "heard of dengue" seem peculiar things to include in the regression, as these might be extended to be a consequence of having dengue rather than a potential risk factor. Perhaps this reflects the fact that the purpose of the regression analysis is not clearly stated. It would be good to have a clear statement about the purpose of the regression and a justification for the choice of variables to include.

We agree with the reviewer that it was confusing to include them in the regression as the main purpose was to look at risk factors. We now remove them from the regression.

14) The supplementary material describes the recruitment strategy: "the study staff identified the house where the most recent wedding had taken place and identified the closest neighbour. They then counted six households in a random direction to identify the first household for the study. To select each additional household for the study, they used the previous household as a starting point and counted six households in a random direction." This seems a little eccentric and not obviously guaranteed to give a random sample. Is there any justification for this choice? Wouldn't numbering households and selecting at random or throwing darts at maps be better? I think this point at least needs to be discussed (with recommendations for future studies) and this aspect of the methods should be move to the Methods section in the main text.

There do not exist detailed maps for each community (nor indeed maps that can tell you reliably where any community is located). While mapping and enumerating households by the teams would be preferable – in reality this is rarely feasible where there can be hundreds of households and the teams only had a short amount of time in each community. The sampling approach has been used frequently by the teams in icddr,b to identify a random geographic point within the community to begin household enrolment. In practice it allows the teams to cover a substantial part of the communities. It is also difficult to think of how such an approach could have brought in a systematic bias, especially given there tend to not be many differences within Bangladeshi communities and over 95% of adults marry in the country. Nevertheless, we agree that the strategy may appear a little unusual and in the revised document we discuss that it would have preferable to enumerate all households but this was not feasible in this study.

We now include the recruitment strategy in the main document. In the Discussion we include: “To randomly select households in a community, we would ideally have used a sampling frame of all households in the community. […] However, in cultural contexts where marriage rates may vary by community or location within a community, this method of choosing a random starting point could produce a biased sample.”

15) "may provide some guidance". Not sure what the intended meaning of this is. What kind of guidance?

This has now been removed.

16) Was any attempt made at assessing the accuracy of the recorded household data?

The interviewers visually confirmed the accuracy of the reported household characteristics through observation where this was possible.

17) The entomological approach to determining the presence or absence of Ae. aegypti is quite superficial. The intensity of surveillance (number of BG traps per community) and duration (time in the field) is too short to arrive at a conclusion of presence/absence. In dengue endemic cities, where Ae. aegypti has a well-documented presence, there can be quite marked spatial heterogeneity in the distribution of Ae. aegypti when measured by BG traps, e.g. some houses can be free of this species for consecutive weeks but in a house 50 metres away they can be caught regularly. In regards to Ae. albopictus prevalence, BG traps set indoors are not the optimal method of determining presence/absence- it would have better to set them outdoors or to use outdoor ovitraps. The authors should qualify their conclusions by recognising that trapping method, intensity, duration and seasonality can all influence the likelihood of Aedes detection and this could change the conclusions of the manuscript.

We feel confident with our approach of determining presence or absence of *Aedes* within our sampled communities. We used BG traps that are very well suited for *Aedes* mosquitoes (including *albopictus* – see Maciel-de-Freitas et al., 2006; Obenauer et al., 2010). Further, we revisited all communities where we did not record *Aedes* populations in the initial visit and in each visit, we conducted 8 households x 24 hours trap-hours of collection. This means that in negative communities, no mosquitoes were trapped over 16 x 24 = 384 hours. We specifically targeted the known *Aedes* season for trapping mosquitoes in communities where the first trapping efforts detected no mosquitoes. We are also confident with the specificity of identifying species, as trained and experienced entomologists were used to speciate the mosquitoes (therefore false instances of presence are very unlikely). Finally, the strong spatial correlation in where each mosquito species were found (see Author response image 1) would be difficult to achieve if the mosquito data were not robust. Nevertheless, in the revised document, we include a discussion that we could have missed some mosquitoes (i.e., there may be some false negatives).

In the Discussion, we include “We used BG sentinel traps that have been shown to be well suited to trapping Aedes mosquitoes 34,35. […] This would mean that Aedes may be more widespread than we found.”

18) Is there any reason uncertainty (in either or all of the data, kriging model and the force of infection model) can't be propagated through to the final burden estimates? Comparing mean estimates with Bhatt et al. to prove that nationally representative surveys are needed probably also needs to consider uncertainty. I think they might have also included tertiary and quaternary infections as well if you want to be comparable.

Thank you for this suggestion. We now include uncertainty in the estimates. In Bhatt et al., the authors rely on fitting models to inapparent:apparent ratios from cohort data that have paired serology – as tertiary (and quaternary) infections will be IgG positive in both samples and IgM responses will be the same as secondary infections, this approach cannot usually detect these infections. A key issue of the Bhatt paper is that it failed to account for population immunity. Hence the high estimates. In many places (India included) they estimate more infections than births/year. In any event we now include tertiary and quaternary infections in the estimates, which gives similar estimates (only a small subset of the population has the right immune history to have these sequences of heterotypic infections) – it moves the total number of estimated infections from 2.32 million to 2.35 million.

We now include in the Results methods on how we calculated the number of infections. In the Results we include “Using a catalytic model to estimate the proportion seropositive by age, we estimated that 1.6% (95% CI: 1.5%-1.7%) of the susceptible population gets infected each year across the four serotypes, equivalent to an average of 2.4 million annual infections (95% CI: 2.2-2.5 million).”

19) Can code for statistical analysis be made available?

We will deposit code for the manuscript on GitHub.

20) "Our findings are in marked contrast to what has been observed with chikungunya in Bangladesh". Can the authors offer any hypothesis as to why this might be the case?

The differences in the chikungunya versus dengue experiences may be due to vector, level of endemicity or behavioral differences. We have changed this section to read “Our findings are in marked contrast to what has been observed with chikungunya in Bangladesh, where women in a community that had a widespread chikungunya epidemic were found to be at significantly increased risk of being infected compared to males, with the increased risk of infection linked to greater time spent in and around the home. These findings suggest that it may be difficult to generalize inferences across arboviruses due to differences in vector species and the frequency of introductions and risk of onwards spread.”

21) Can the authors provide at least one concrete example of such a survey could lead to better decision-making about vaccination?

We now include the following: “For Bangladesh, our findings suggest that any vaccine rollout should be concentrated to the urban areas of Dhaka, Chittagong and Khulna. […] Therefore, any rollout will require the screening of individuals for presence of antibodies before vaccination to avoid placing large numbers of individuals at risk for more severe disease manifestations.”

22) It was felt that some of the statements about findings showing that lack of spread of aegypti is the reason behind heterogeneities in dengue transmission in Bangladesh were not fully justified in light of the known limitations of entomological surveying

As discussed in the response to comment 15 above, we feel confident in our estimates of mosquito presence. Even for *Aedes albopictus*, the BG trap has been shown to be very good (see https://doi.org/10.1603/EN09322). We would not observe such clear spatial correlation in where the mosquitoes were found (and not found) if the observations were not robust. Nevertheless, we agree that it is possible that we have missed the mosquitoes in a small number of communities (i.e. false negatives). We include this point in the Discussion. The effect of mosquito distribution has also been downplayed throughout the document.

We now include in the Discussion: “We used BG sentinel traps that have been shown to be well suited to trapping *Aedes* mosquitoes (Maciel-de-Freitas, Eiras and Lourenço, 2006; Obenauer et al., 2010). […] This would mean that *Aedes* may be more widespread than we found.”

23) The authors correctly point to the possibility that the Panbio-based seroprevalence survey might have reflected past JEV exposure and cite the low case JE incidence to suggest dengue is the primary culprit for the seroloprevalence.. This might be true, but JEV is notoriously difficult to diagnose in the absence of laboratory testing and there would almost certainly be under-reporting of cases in Bangladesh. Having a random subset of samples tested by DENV/JEV PRNT50 assay, regarded as the most specific assay of DENV serostatus, could have helped clarify this point. Can the authors do this readily perhaps on a sample of 100 or so (or use Luminex as a (less preferred) option? Though not essential, it was felt this would improve the quality of the manuscript if it could be done within 2 months. If it can't be done the Discussion needs to be qualified accordingly.

We agree that additional PRNT testing for JE and dengue on some of our samples would be useful, but we are unable to arrange this within a few months. The reviewer is correct that JEV is difficult to diagnose, but our estimation that past JEV exposure is very low among the general population is based upon laboratory confirmed patients who had samples tested at the US CDC as part of a long-running hospital-based surveillance program. These data showed that approximately half of all cases diagnosed are adults, suggesting a low force of infection. A community-based study that a combined population-based door to door survey looking for people who had ever had an illness compatible with encephalitis with the estimated prevalence of JEV among hospitalized encephalitis patients with confirmed infection estimated only 2.7 cases/100,000 population in that part of the country. The low level of JE in Bangladesh is not surprising, as pigs (the main reservoir for JE) are uncommon in the predominantly Muslim country.

Finally, laboratory diagnosed JEV cases in Bangladesh have all been rural residents and the levels for dengue seropositivity we found was low in rural environments. This means the level of false positives due to JE exposure is likely to be minimal. Note that Luminex testing does not resolve the cross-reactivity problems completely and neutralization testing would still be necessary. We have clarified this in the Discussion.

The Discussion now includes: “Cross-reactivity of antibodies is a problem for all seroprevalence studies, especially with flaviviruses. […] As we estimated only low levels of dengue seropositivity in rural communities, the number of false positives from Japanese encephalitis cross-reactivity is likely to be small.”

24) Materials and methods: when were interviews conducted in relation to the dengue season? Could this have introduced recall bias of "whether diagnosed with dengue"?

The interviews were conducted towards the end of the dengue season. However, as the probability of infection from any particular season is small, any infections are likely to be several years old and it is very possible that they were not and falsely recalled. The purpose of the question was to assess whether asking somebody about their dengue history was at all informative of their true status.

25) Was there any evidence that travel in the last 7 days was a reasonable proxy for long term travel history?

We had no external validation of this measure, although (on a population scale at least), it seems likely that recent travel is a marker of frequently leaving the community. We include this in the limitations in the revised document.

We include: “We used the time period since individuals last left the community as a marker of travel. While this is likely to broadly capture trends in mobility, it remains a crude marker and more detailed measures of movement (from e.g., movement diaries, GPS monitors) would help provide a more detailed understanding of how people move.”

26) "Our finding of a negative correlation between Ae. aegypti and Ae. albopictus presence is consistent with competition between the two species" – or is just evidence that they have different environmental niches?

We agree that this is an alternative explanation and have added: “Our finding of a negative correlation between *Ae. aegypti* and *Ae. albopictus* presence is consistent with the species occupying different environmental niches or even competition between the two species.”